# CentroidKV: Efficient Long-Context LLM Inference via KV Cache Clustering

Jie Hu[1,2*], Shengnan Wang[2*] Yutong He[1], Ping Gong[3], Jiawei Yi[3], Juncheng Zhang[3],
Youhui Bai[2], Renhai Chen[2], Gong Zhang[2], Cheng Li[3], Kun Yuan[1†]
[1]*Peking University*   [2]*Huawei Technologies*   [3]*University of Science and Technology of China*

Reviewed on OpenReview: `https://openreview.net/forum?id=T3EeupQhGj`

## Abstract

Large language models (LLMs) with extended context windows have become increasingly prevalent for tackling complex tasks. However, the substantial Key-Value (KV) cache required for long-context LLMs poses significant deployment challenges. Existing approaches either discard potentially critical information needed for future generations or offer limited efficiency gains due to high computational overhead. In this paper, we introduce *CentroidKV*, a simple yet effective framework for online KV cache clustering. Our approach is based on the observation that key states exhibit high similarity along the sequence dimension. To enable efficient clustering, we divide the sequence into chunks and propose *Chunked Soft Matching*, which employs an alternating partition strategy within each chunk and identifies clusters based on similarity. CentroidKV then merges the KV cache within each cluster into a single centroid. Additionally, we provide a theoretical analysis of the computational complexity and the optimality of the intra-chunk partitioning strategy. Extensive experiments across various models and long-context benchmarks demonstrate that CentroidKV achieves up to 75% reduction in KV cache memory usage while maintaining comparable model performance. Moreover, with minimal computational overhead, CentroidKV accelerates the decoding stage of inference by up to $1.92\times$ and increases the serving throughput by up to $4\times$.

## 1 Introduction

With the increasing demand to tackle a diverse range of complex real-world applications, such as multi-round dialogues, Large Language Models (LLMs) have been capable of supporting context windows of up to 1M tokens (Achiam et al., 2023; Touvron et al., 2023; Team et al., 2024). However, deploying LLMs in long-context scenarios introduces substantial challenges, particularly related to the Key-Value (KV) cache. The KV cache stores the keys and values of all preceding tokens to avoid re-computation, and its memory requirements scale linearly with the context length. Due to the auto-regressive nature of LLMs, generating each token necessitates accessing the entire KV cache, making it a significant bottleneck for both inference latency and throughput. Moreover, the large size of KV cache imposes considerable demands on memory capacity, further complicating deployment.

To address these challenges, there is a pressing need for effective methods to reduce the KV cache size. Existing studies have explored this problem from multiple perspectives, including KV cache eviction (Zhang et al., 2023; Xiao et al., 2023; Ge et al., 2023; Li et al., 2024; Liu et al., 2024b; Yang et al., 2024; Cai et al., 2024), KV cache merging (Zhang et al.; Wang et al., 2024; Wan et al., 2024), quantization (Hooper et al., 2024; Liu et al., 2024c), and channel pruning (Xu et al., 2024), among others. By leveraging the inherent sparsity of the attention score matrix (Zhang et al., 2023), various methods have been proposed to reduce redundancy along the sequence length dimension. However, these approaches exhibit notable limitations. KV

---

[*]Equal contribution.
[†]Corresponding author.

cache eviction, which discards less critical tokens based on historical attention scores, often leads to significant performance degradation. This occurs because tokens deemed unimportant in the current context may become crucial for future generations, especially in long-context scenarios. To improve the model performance after eviction, recent works (Yang et al., 2024; Cai et al., 2024; Tang et al., 2024; Feng et al., 2024; Xiao et al., 2024; Shi et al., 2024) allocate different KV cache budget across layers and attention heads. Additionally, subsequent methods such as (Zhang et al.) merge the tokens to be evicted with the tokens to be retained. However, existing methods typically rely on preliminary token eviction to define merging sets, which often forces semantically distant tokens to be grouped together, resulting in centroid bias and information loss, ultimately degrading model performance. In contrast, our approach aims to compress the KV cache by clustering them into centroids based on the intrinsic similarity.

Our approach is inspired by a key observation: key states exhibit high similarity along the sequence dimension, as illustrated in Figure 2. This insight motivates us to cluster the KV cache by identifying merging sets solely based on token similarity. However, efficiently and accurately performing KV cache clustering online remains challenging, particularly given the long sequence lengths involved.

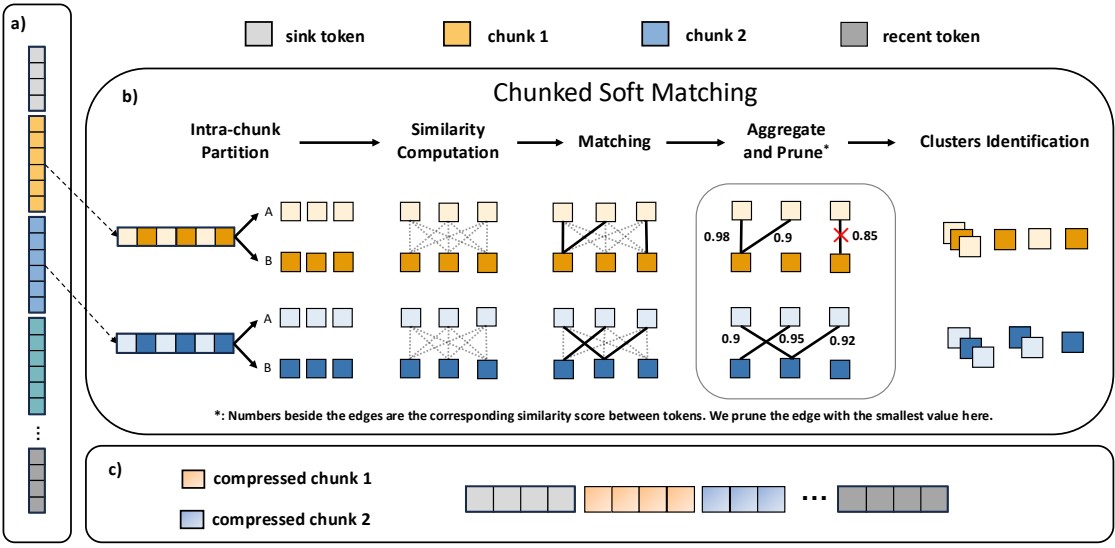

Figure 1: An overview of CentroidKV: a) Divide sequences into chunks; b) Chunked Soft Matching to identify clusters; c) KV cache compression after clustering.

In this paper, we introduce *CentroidKV*, a simple yet effective online KV cache clustering framework that improves the inference efficiency of LLMs in long-context scenarios. The core innovation of CentroidKV is the *Chunked Soft Matching* algorithm, which enables efficient KV cache clustering with a favorable accuracy–efficiency trade-off. Inspired by the Bipartite Soft Matching algorithm (Bolya et al., 2022) for Vision Transformers (Dosovitskiy, 2020), we are the first to extend this idea to the context of KV cache compression. Given the cache budget, one clustering round is executed as follows. The framework begins by dividing the sequence into chunks. Based on the observed correlation between token similarity and positional distance, we further partition each chunk in an alternating manner and theoretically prove the optimality of this intra-chunk partitioning strategy. Subsequently, *Chunked Soft Matching* identifies clusters by locating highly similar token pairs across all chunks. Finally, the corresponding keys and values within each cluster are merged into a single centroid. As the decoding process advances, CentroidKV calls the clustering process if the cache size exceeds the budget.

We conduct extensive experiments on popular models to evaluate both the effectiveness and efficiency of CentroidKV. The results demonstrate that CentroidKV can reduce the KV cache memory usage by up to 75% while maintaining comparable model performance. Furthermore, by dynamically clustering to preserve contextual information, CentroidKV outperforms baselines under a limited cache budget. Additionally, due

to its simplicity and efficiency, CentroidKV accelerates the decoding stage of LLM inference by up to 1.92×
and increases the throughput by up to 4× compared to full KV cache.

Our contributions are summarized as follows.

- We introduce *CentroidKV*, a simple yet effective framework for online KV cache clustering. The core
innovation is a novel clustering algorithm, termed *Chunked Soft Matching*.

- CentroidKV is a lightweight, plug-and-play solution to improve LLM inference efficiency. We
theoretically analyze its computational complexity and formally prove the optimality of the intra-
chunk partitioning strategy based on the observed similarity patterns.

- CentroidKV achieves up to a 75% reduction in KV cache memory usage with minimal impact on
model performance. Additionally, it accelerates the decoding stage by up to 1.92× and increases the
throughput by up to 4×.

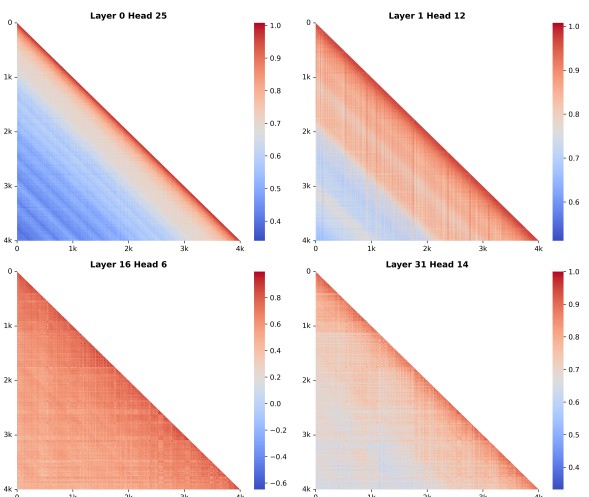

Figure 2: The cosine similarity maps of key states
across various layers and heads.

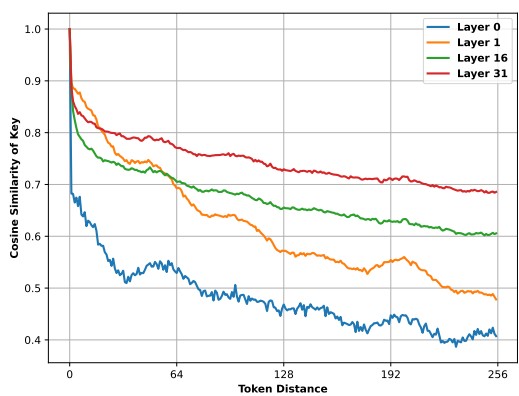

Figure 3: The correlation between token distance and
the cosine similarity of key states.

## 2    Related Work

**KV Cache Compression.**    Eviction-based approaches aim to maintain a fixed-size KV cache during
decoding by selectively retaining informative tokens. H2O (Zhang et al., 2023) retains a limited budget of KV
cache by greedily discarding unimportant tokens based on the accumulated attention score. StreamingLLM
(Xiao et al., 2023) preserves the initial few tokens along with the recent tokens, based on the identification
of attention sinks. SnapKV (Li et al., 2024) selects important tokens for each attention head based on
the observation window of prompts. FastGen (Ge et al., 2023) conducts profiling for attention heads and
dynamically evicts tokens based on different attention patterns. Despite their efficiency, these methods
suffer from performance degradation in the long context scenario, since the information of the discarded
token will be lost permanently. To mitigate this limitation, subsequent approaches (Zhang et al.; Wan et al.,
2024) introduce compensation mechanisms that merge evicted tokens into the remaining cache. However,
their reliance on preliminary eviction still constrains their ability to preserve full contextual information.
ClusterKV (Liu et al., 2024a) employs K-means clustering to enhance the retrieval of salient tokens; however,
the high computational overhead of iterative clustering procedures limits its practicality for latency-sensitive
online inference scenarios. Another line of work, such as PyramidKV (Cai et al., 2024) and PyramidInfer
(Yang et al., 2024), focuses on optimizing cache budget allocation across layers to improve overall model
accuracy.

**Bipartite Soft Matching.** Introduced by ToMe (Bolya et al., 2022), Bipartite Soft Matching (BSM) is an algorithm designed to merge tokens in Vision Transformers (ViT) (Dosovitskiy, 2020), which is as fast as token pruning. Recent studies (Bolya & Hoffman, 2023; Kim et al., 2024; Tran et al., 2024) have extended BSM to enhance the efficiency of ViTs and Stable Diffusion (Rombach et al., 2022) without compromising model performance.

## 3 Preliminary

In this section, we first give a basic preliminary about the KV cache clustering in LLM inference. For simplicity, we focus on a single attention head within a specific layer. LLM inference consists of two stages: pre-filling and decoding. During the pre-filling stage, the model generates the first token and initializes the KV cache, which stores the key and value states of the prompt as $K \in \mathbb{R}^{n \times d}$ and $V \in \mathbb{R}^{n \times d}$, respectively. In the decoding stage, for the given states of the current input $q, k, v \in \mathbb{R}^{1 \times d}$, KV cache is updated as $K = [K, k]$, $V = [V, v]$. The vanilla attention output is then computed as follows:

$$\text{Attn}(q, K, V) = \text{softmax}\left(\frac{qK^T}{\sqrt{d_k}}\right)V = \frac{\sum_{i=1}^n \exp\left(\frac{q^T k_i}{\sqrt{d_k}}\right) v_i}{\sum_{i=1}^n \exp\left(\frac{q^T k_i}{\sqrt{d_k}}\right)}. \tag{1}$$

For clearer derivation, we decompose KV cache into individual tokens, represented as $K = [k_1, \ldots, k_n]$ and $V = [v_1, \ldots, v_n]$, where each $k_i \in \mathbb{R}^d$ and $v_i \in \mathbb{R}^d$. In our approach, we utilize the cosine similarity as the distance metric between key states:

$$\cos(k_i, k_j) = \frac{k_i \cdot k_j}{\|k_i\| \|k_j\|} \tag{2}$$

Assume that tokens are grouped into clusters based on the similarity of their key states. The cluster centers are denoted as $\hat{K} = [\hat{k}_1, \ldots, \hat{k}_C]$, where $C$ represents the number of clusters. The number of tokens in each cluster is denoted by $N = [n_1, \ldots, n_C]$, with each cluster center $\hat{k}_t$ associated with a set of tokens $[\hat{k}_{1,t}, \ldots, \hat{k}_{n_t,t}]$, corresponding to the tokens in the $t$-th cluster. Here, $n_t$ is referred to as the cluster degree.

By approximating the attention output using the cluster centers in place of the original key states, the resulting attention output is as follows:

$$\text{Attn}(q, K, V) = \frac{\sum_{t=1}^C \sum_{i=1}^{n_t} \exp\left(\frac{q^T k_{i,t}}{\sqrt{d_k}}\right) v_{i,t}}{\sum_{t=1}^C \sum_{i=1}^{n_t} \exp\left(\frac{q^T k_{i,t}}{\sqrt{d_k}}\right)}$$

$$\approx \frac{\sum_{t=1}^C \sum_{i=1}^{n_t} \exp\left(\frac{q^T \hat{k}_t}{\sqrt{d_k}}\right) v_{i,t}}{\sum_{t=1}^C \sum_{i=1}^{n_t} \exp\left(\frac{q^T \hat{k}_t}{\sqrt{d_k}}\right)}$$

$$= \frac{\sum_{t=1}^C \left[\exp\left(\frac{q^T \hat{k}_t}{\sqrt{d_k}}\right) \sum_{i=1}^{n_t} v_{i,t}\right]}{\sum_{t=1}^C n_t \exp\left(\frac{q^T \hat{k}_t}{\sqrt{d_k}}\right)}.$$

By merging the value states corresponding to the same indices as the key states, namely $\hat{v}_t = \frac{\sum_{i=1}^{n_t} v_{i,t}}{n_t}$, we formulate the approximate attention output, denoted as AppAttn, as follows:

$$\text{Attn}(q, K, V) \approx \frac{\sum_{t=1}^C n_t \exp\left(\frac{q^T \hat{k}_t}{\sqrt{d_k}}\right) \hat{v}_t}{\sum_{t=1}^C n_t \exp\left(\frac{q^T \hat{k}_t}{\sqrt{d_k}}\right)}$$

$$= \frac{\sum_{t=1}^C \exp\left(\frac{q^T \hat{k}_t}{\sqrt{d_k}} + \log n_t\right) \hat{v}_t}{\sum_{t=1}^C \exp\left(\frac{q^T \hat{k}_t}{\sqrt{d_k}} + \log n_t\right)}$$

$$\triangleq \text{AppAttn}(q, \hat{K}, \hat{V}, N), \tag{3}$$

where $\hat{V} = [\hat{v}_1, \ldots, \hat{v}_C]$. Equation (3) demonstrates that the approximate attention after clustering aligns with the vanilla attention in Equation (1) when each token is treated as a separate cluster. To minimize output error, it is essential that the key states within each cluster exhibit high similarity. Fortunately, the following observations further support the viability of this approach.

## 4 Observations

In this section, we present several empirical observations that motivate our approach.

**Observation 1. Key states exhibit high, localized similarity along the sequence dimension.**

Using the Llama-2-7B-32K model (Together, 2023), we randomly sample sequences of length 4K from the WikiText-2 (Merity et al., 2016) dataset and perform zero-shot inference. The cosine similarity maps of the key states are visualized in Figure 2. We observe that key states exhibit high cosine similarity between tokens across different layers and heads. Notably, tokens with high similarity tend to cluster within localized regions. These findings align with previous work (Wang et al., 2024).

This observation suggests that KV cache clustering can be leveraged for efficient inference without compromising accuracy. Furthermore, the observed localized similarity motivates us to subsequently enhance clustering efficiency by identifying similar tokens within local regions, rather than considering the entire sequence.

**Observation 2. As token distance increases, the cosine similarity of key states generally decreases monotonically and follows a convex trend.**

Building on the observed localized similarity, we further investigate the correlation between token distance and the cosine similarity of key states. We randomly sample multiple tokens within the sequence, define the local region as 256 tokens, and compute the average similarity across samples and attention heads. As illustrated in Figure 3, we find that as token distance increases, the cosine similarity between key states generally follows a monotonically decreasing trend. Furthermore, this correlation appears to be convex with respect to the distance.

This observation motivates our framework design of a highly efficient clustering algorithm that minimizes the computational complexity of identifying similar token sets. Additionally, it provides empirical support for the theoretical analysis in Appendix A which proves the optimality of the partitioning strategy in our framework.

## 5 Method

In this section, we introduce *CentroidKV*, a simple yet effective framework for KV cache clustering designed to enhance the efficiency of long-context LLM inference. The framework comprises three key steps. First, the sequence is divided into chunks while preserving the attention sinks and recent tokens. Second, we propose a novel KV cache clustering algorithm, termed *Chunked Soft Matching*, which employs an alternating partition strategy within each chunk and identifies clustering sets by finding highly similar token pairs across all chunks. Finally, the key and value states are merged based on the identified clusters, yielding a compressed KV cache for subsequent generations. An overview of CentroidKV is depicted in Figure 1.

### 5.1 Overall Inference Pipeline

The overall pipeline of LLM inference integrated with CentroidKV is illustrated in Algorithm 1. The cache budget $B$ is determined by the prompt length $n$ and the cache ratio $R$. In the pre-filling stage, after computing the attention output by Flash Attention (Dao et al., 2022), CentroidKV is invoked to conduct the first clustering if the KV cache size exceeds the budget, which compresses the KV cache to centroids and records the cluster degree. In the decoding stage, the KV cache grows by one token at each step, causing the memory budget to be exceeded continuously. To amortize the compression overhead and maintain inference efficiency, CentroidKV is invoked periodically every $g$ decoding steps.

As illustrated in Figure 1, CentroidKV primarily employs the Chunked Soft Matching algorithm to compress the KV cache. The compression ratio $r$ in Algorithm 1 governs the proportion of pruned edges in Figure 1.

---

**Algorithm 1** Inference Pipeline with CentroidKV

---

**Require:** cache ratio $R$, compression ratio $r$, maximum decoding length $\Gamma$, attention sink $n_1$, recent budget $n_2$, interval step $g$, chunk size $c$

1: **Pre-filling:** $Q, K, V \in \mathbb{R}^{n \times d}$
2: Cache length $s = n$, cluster degree $N = [1] \cdot n$, cache budget $B = R \cdot n$        ▷ Initialization
3: $O = \text{FlashAttn}(Q, K, V)$
4: **while** $s > B$ **do**
5:      $K, V, N, s = \text{CentroidKV}(K, V, N, s, n_1, n_2, r, c)$      ▷ First clustering after prefilling
6: **end while**
7: **Decoding:** $q, k, v \in \mathbb{R}^{1 \times d}$
8: **for** $i = 1, \ldots, \Gamma - 1$ **do**
9:      $K = [K, k]$, $V = [V, v]$, $N = [N, 1]$, $s = s + 1$      ▷ Update the cache centroids
10:      $O = \text{softmax}(qK^T / \sqrt{d} + \log N) \cdot V$      ▷ Equation 3
11:      **if** $s \geq B + g$ **then**
12:         $K, V, N, s = \text{CentroidKV}(K, V, N, s, n_1, n_2, r, c)$      ▷ Clustering every $g$ decoding steps
13:      **end if**
14: **end for**

---

Table 1: Computational complexity of distance matrix. $n$ is the sequence length and $d$ is hidden dimension. $k$ and $i$ of K-Means refer to number of cluster centers and iterations. $c$ of Chunked Soft Matching (CSM) refers to chunk size, which is relatively small to $n$.

| Mesh | K-Means | BSM | CSM |
|------|---------|-----|-----|
| $n^2 d$ | $inkd$ | $\frac{1}{4}n^2 d$ | $\frac{1}{4}ncd$ |

Since Bipartite Soft Matching can reduce the cache size by at most half in a single pass, multiple rounds of clustering are performed to progressively compress the KV cache until it meets the predefined budget.

## 5.2 Chunked Soft Matching

Based on Observation 1 in Section 4, CentroidKV begins by dividing the key states into chunks, as illustrated in Figure 1. The localized similarity of key states ensures that this approach will not cause a significant loss of accuracy. We keep the attention sink and recent tokens before partitioning, regarding their importance for model performance (Zhang et al., 2023; Xiao et al., 2023).

Inspired by the Bipartite Soft Matching (BSM) algorithm introduced by (Bolya et al., 2022) for token merging in the transformer block of Vision Transformers (ViT) (Dosovitskiy, 2020), we propose Chunked Soft Matching (CSM) for KV cache clustering in the second step of CentroidKV. BSM begins by partitioning the input tokens into two distinct sets, $A$ and $B$. Next, for each token in set $A$, an edge is drawn to its most similar token in set $B$. Among these edges, only the top-ranked similar connections are retained. Tokens that remain connected through these edges are then merged into a cluster, while others are left unchanged. Finally, the two sets, $A$ and $B$, are concatenated to form the output, incorporating both merged and unchanged tokens.

However, directly applying BSM for KV cache clustering poses two significant challenges. The first challenge arises from the large sequence dimension in long-context scenarios, which leads to inefficiencies in the matching process. The second challenge involves optimally partitioning the sequence into two sets, A and B, in a way that minimizes the impact on model accuracy.

CSM addresses the first challenge through a preceding chunking step, which directly improves computational efficiency. While there are alternative approaches for KV cache clustering, such as K-means, they typically require multiple iterative updates to establish stable clusters, leading to substantial computational overhead. We summarize the computational complexity of representative methods in Table 1. In particular, the "Mesh" method computes pairwise cosine similarity across all tokens, incurring significant cost. In contrast,

CSM achieves the lowest complexity by performing a single-pass clustering procedure with minimal token interactions.

Building on Observation 2 in Section 4, we address the second challenge by partitioning each chunk into two sets, $A$ and $B$, in an alternating manner. The core idea of CSM is to assign highly similar states to different sets, ensuring that token pairs with high similarity are placed between sets, rather than within them. Furthermore, based on the observed monotonically decreasing and convex trend, we rigorously demonstrate the theoretical optimality of this intra-chunk partitioning strategy in Appendix A. After computing the similarities, CSM aggregates the edges from all chunks and produces candidate matched pairs for KV cache merging.

### 5.3 KV Cache Compression

After candidate clusters are determined, CentroidKV performs selective merging to construct the compressed KV cache. Importantly, as shown in Figure 1, not all matched pairs are merged. Instead, we rank all candidate matches by similarity and only merge the top fraction controlled by a compression ratio $r$. Concretely, $r < 1.0$ retains only the top-$r$ most similar matches, discarding lower-confidence ones. This design avoids noisy aggregations from weakly similar tokens. As shown in Algorithm 1, clustering is applied over multiple rounds to satisfy the cache budget. To further improve clustering quality across rounds, we adopt a progressively decreasing schedule for $r$, initialized at $r_{\text{init}}$ and linearly annealed with decay rate $\delta_r$:

$$r = r_{\text{init}} - j \cdot \delta_r \tag{4}$$

where $j$ is the clustering round index. This schedule makes merging increasingly selective over time, preserving high-confidence structure in later rounds.

For each merged cluster, we maintain a degree counter $n_t$ to track its cumulative contribution across rounds. Both key and value states are aggregated using degree-weighted averaging. Specifically, for a cluster $k_1, k_2, \ldots, k_t$ with corresponding degrees $n_1, n_2, \ldots, n_t$, the centroid is computed as:

$$\hat{k} = \frac{n_1 k_1 + n_2 k_2 + \cdots + n_t k_t}{n_1 + n_2 + \cdots + n_t} \tag{5}$$

Value states are merged analogously using the same degree-based weighting.

Finally, the resulting centroids are concatenated with the preserved attention sink and recent tokens to form the compressed KV cache for subsequent decoding steps.

## 6 Experiments

In this section, we conduct comprehensive experiments to evaluate the effectiveness and efficiency of CentroidKV, followed by the ablation study on the framework design.

### 6.1 Experimental Settings

**Models and baselines.** We employ two long-context models: Llama-3.1-8B-Instruct (Meta, 2024) and Mistral-7B-Instruct-v0.2 (Jiang et al., 2023), serving as the backbone LLMs. We compare CentroidKV against state-of-the-art KV cache compression methods, including StreamingLLM (Xiao et al., 2023), SnapKV (Li et al., 2024) and PyramidKV (Cai et al., 2024).

**Datasets.** We evaluate CentroidKV using the widely recognized benchmarks: RULER (Hsieh et al., 2024) and LongBench (Bai et al., 2023). As the variation of the Needle-in-a-Haystack test (Kamradt, 2024), RULER includes 13 long-sequence tasks designed to assess the long-context understanding capabilities of LLMs. Additionally, LongBench includes tasks covering various application scenarios: single-document QA, multi-document QA, summarization, few-shot learning, synthetic tasks, and code completion.

Table 2: Detailed comparison of RULER benchmark datasets across various budgets on Llama-3.1-8B-Instruct.

| Budget | Method | CWE | FWE | MK-NIAH-1 | MK-NIAH-2 | MK-NIAH-3 | MQ-NIAH | MV-NIAH | S-NIAH-1 | S-NIAH-2 | S-NIAH-3 | QA-1 | QA-2 | VT |
|---|---|---|---|---|---|---|---|---|---|---|---|---|---|---|
| 100% | Full | 99.31 | 95.80 | 100.00 | 100.00 | 100.00 | 100.00 | 100.00 | 100.00 | 100.00 | 100.00 | 88.24 | 56.67 | 99.77 |
| 75% | StreamingLLM | **99.71** | 93.99 | 84.62 | 80.68 | **75.00** | 76.52 | 77.31 | 79.51 | 77.06 | 71.91 | **87.25** | 51.11 | 94.32 |
| | SnapKV | 99.02 | 95.20 | 98.08 | 81.82 | 56.82 | 95.45 | 89.58 | 98.36 | 100.00 | 10.11 | 84.31 | 51.11 | 92.73 |
| | PyramidKV | 99.51 | 93.69 | 98.08 | 85.23 | 53.41 | 98.23 | 97.92 | 99.18 | 100.00 | 10.11 | 86.27 | 47.78 | 95.91 |
| | CentroidKV | 99.12 | **95.20** | **100.00** | **100.00** | 60.23 | **99.75** | **98.61** | **100.00** | **100.00** | 87.64 | 84.31 | **54.44** | **99.32** |
| 50% | StreamingLLM | 58.04 | 91.59 | 54.81 | 46.59 | **51.14** | 53.54 | 52.78 | 48.36 | 42.20 | **53.93** | **87.25** | 47.78 | 73.64 |
| | SnapKV | **98.53** | 92.49 | 91.35 | 47.73 | 22.73 | 78.03 | 73.38 | 95.08 | 93.58 | 2.25 | 73.53 | 43.33 | 80.68 |
| | PyramidKV | 88.63 | 90.39 | 85.58 | 57.95 | 18.18 | 76.01 | 77.78 | 96.72 | 97.25 | 1.12 | 73.53 | 43.33 | 81.82 |
| | CentroidKV | 97.45 | **93.39** | **99.04** | **81.82** | 4.55 | **97.47** | **92.13** | **100.00** | **100.00** | 37.08 | 78.43 | **52.22** | **96.59** |
| 25% | StreamingLLM | 10.29 | **93.39** | 32.69 | 22.73 | **21.59** | 29.04 | 26.85 | 27.87 | 19.27 | **28.09** | 89.22 | 38.89 | 43.64 |
| | SnapKV | 86.08 | 85.59 | 33.65 | 10.23 | 1.14 | 24.75 | 24.07 | 80.33 | 53.21 | 1.12 | 55.88 | 27.78 | 59.32 |
| | PyramidKV | **86.27** | 85.59 | 33.65 | 10.23 | 1.14 | 23.23 | 24.07 | 80.33 | 53.21 | 1.12 | 55.88 | 26.67 | 59.32 |
| | CentroidKV | 85.10 | 87.69 | **87.50** | **23.86** | 0.00 | **77.27** | **67.36** | **100.00** | **96.33** | 3.37 | 36.27 | 30.00 | **88.64** |

Table 3: Detailed comparison of RULER benchmark datasets across various budgets on Mistral-7B-Instruct.

| Budget | Method | CWE | FWE | MK-NIAH-1 | MK-NIAH-2 | MK-NIAH-3 | MQ-NIAH | MV-NIAH | S-NIAH-1 | S-NIAH-2 | S-NIAH-3 | QA-1 | QA-2 | VT |
|---|---|---|---|---|---|---|---|---|---|---|---|---|---|---|
| 100% | Full | 95.88 | 92.79 | 99.04 | 100.00 | 93.18 | 98.48 | 87.27 | 100.00 | 99.08 | 98.88 | 86.27 | 62.22 | 98.86 |
| 75% | StreamingLLM | 92.75 | **91.89** | 84.62 | 80.68 | **68.18** | 65.91 | **73.38** | 79.51 | **77.06** | 70.79 | **86.27** | 56.67 | 52.73 |
| | SnapKV | 95.00 | 90.99 | 38.46 | 76.14 | 67.05 | 32.32 | 25.00 | 87.70 | 67.89 | 6.74 | 85.29 | 58.89 | 27.50 |
| | PyramidKV | **95.98** | 90.99 | 47.12 | 76.14 | 57.95 | 38.13 | 30.79 | 93.44 | 69.72 | 6.74 | 84.31 | **62.22** | 55.23 |
| | CentroidKV | 94.02 | 90.69 | **87.50** | **97.73** | 54.55 | **79.80** | 65.97 | **100.00** | 71.56 | **96.63** | 83.33 | 53.33 | **98.64** |
| 50% | StreamingLLM | 90.98 | **89.79** | 54.81 | 46.59 | **40.91** | 33.08 | **52.08** | 46.72 | **42.20** | 52.81 | **87.25** | 47.78 | 50.45 |
| | SnapKV | **94.31** | 88.89 | 23.08 | 44.32 | 17.05 | 15.40 | 14.58 | 78.69 | 22.02 | 2.25 | 76.47 | 51.11 | 18.64 |
| | PyramidKV | 79.80 | 86.79 | 20.19 | 39.77 | 9.09 | 13.64 | 13.89 | 86.89 | 15.60 | 1.12 | 70.59 | **51.11** | 29.09 |
| | CentroidKV | 89.31 | 88.29 | **57.69** | **56.82** | 2.27 | **34.09** | 29.86 | **100.00** | 36.70 | **53.93** | 71.57 | 46.67 | **97.27** |
| 25% | StreamingLLM | 60.20 | **90.99** | 32.69 | 21.59 | **18.18** | 16.67 | **26.16** | 27.87 | **19.27** | 28.09 | 88.24 | 40.00 | 19.32 |
| | SnapKV | **87.35** | 84.08 | 14.42 | 14.77 | 0.00 | 13.89 | 14.35 | 69.67 | 11.01 | 1.12 | 60.78 | 38.89 | 15.68 |
| | PyramidKV | 58.92 | 84.38 | 14.42 | 14.77 | 0.00 | 13.38 | 14.35 | 69.67 | 11.01 | 1.12 | 60.78 | 38.89 | 15.91 |
| | CentroidKV | 76.57 | 74.77 | 17.31 | 11.36 | 0.00 | 5.30 | 9.49 | **100.00** | 15.60 | 3.37 | 51.96 | 34.44 | **94.09** |

**Implementation Details.** All experiments are conducted on the KVPress codebase (Devoto et al., 2025), built upon HuggingFace Transformers (Wolf, 2019). Following KVPress, each input is divided into context and question segments. In our evaluation protocol, only the context is available during compression, while the question is introduced afterward for answer generation. This setup more faithfully reflects real-world deployment, where future queries are unknown at compression time, and therefore constitutes a more challenging and realistic setting.

Unless otherwise specified, CentroidKV is configured with 16 attention sinks and retains the 64 most recent tokens. The initial compression ratio is set to $r_{\text{init}} = 0.8$ with a decay rate of $\delta_r = 0.2$. The default chunk size is 256, and all computations are performed in bfloat16 precision. For experiments on RULER, the default context length is 4K tokens unless stated otherwise. Baseline methods are implemented following the configurations reported in their respective papers. Additional details are provided in Appendix B. All experiments are conducted on high-performance GPUs delivering over 100 TFLOPS of compute.

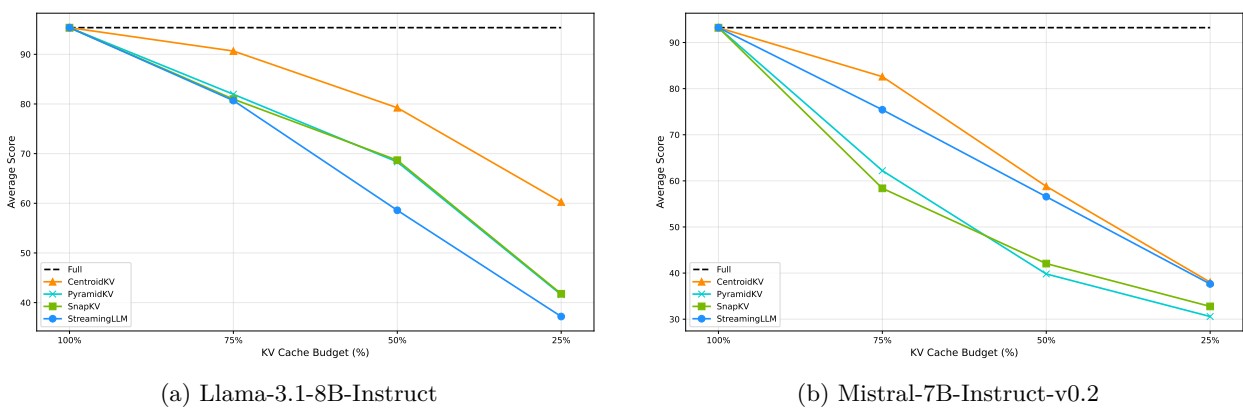

(a) Llama-3.1-8B-Instruct

(b) Mistral-7B-Instruct-v0.2

Figure 4: Average score comparison of RULER benchmark across different cache budgets.

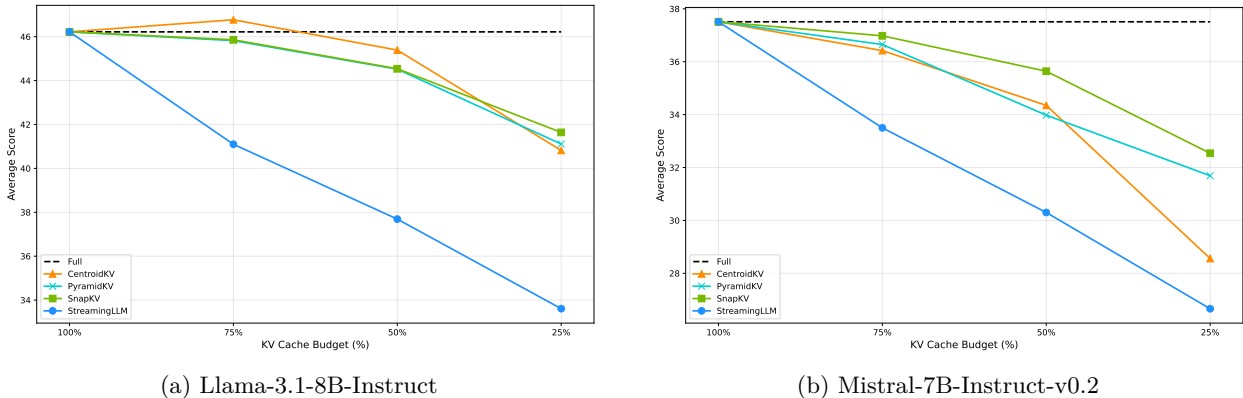

(a) Llama-3.1-8B-Instruct

(b) Mistral-7B-Instruct-v0.2

Figure 5: Average score comparison of LongBench datasets across different cache budgets.

Table 4: Task-average scores of LongBench across various budgets on Mistral-7B-Instruct.

| Budget | Method | Single-Doc QA | Multi-Doc QA | Summarization | Few-shot | Synthetic | Code |
|---|---|---|---|---|---|---|---|
| 100% | Full | 33.00 | 24.54 | 27.75 | 55.18 | 38.12 | 51.25 |
| 75% | StreamingLLM | 25.92 | 22.96 | 26.80 | 51.73 | 28.20 | 48.72 |
| | SnapKV | 31.23 | 23.92 | 27.23 | 54.55 | 39.01 | 51.41 |
| | PyramidKV | 30.71 | 23.92 | 27.19 | 53.36 | **39.32** | 51.10 |
| | CentroidKV | **31.34** | **24.04** | **27.54** | **54.95** | 33.15 | **51.45** |
| 50% | StreamingLLM | 22.92 | 19.52 | 25.90 | 48.87 | 18.27 | 48.30 |
| | SnapKV | 26.91 | **22.24** | 25.96 | **54.34** | **39.23** | **51.70** |
| | PyramidKV | 25.70 | 18.17 | 25.23 | 52.55 | 38.70 | 50.73 |
| | CentroidKV | **29.63** | 20.91 | **26.37** | 51.94 | 31.05 | 50.49 |
| 25% | StreamingLLM | 19.05 | 15.95 | 23.94 | 44.10 | 10.79 | 47.96 |
| | SnapKV | 21.54 | 16.34 | 24.01 | **53.29** | **36.36** | **51.16** |
| | PyramidKV | 20.71 | 15.53 | 23.61 | 51.94 | 35.08 | 50.77 |
| | CentroidKV | **23.02** | **17.09** | **24.47** | 42.50 | 19.13 | 48.72 |

## 6.2 Accuracy Evaluation

We evaluate accuracy under varying KV cache budgets on RULER and LongBench, reporting results at 25%–75% of the full cache to characterize the accuracy–compression trade-off.

**RULER.** Figure 4 shows average performance over 13 RULER datasets. CentroidKV achieves the strongest overall performance on both Llama-3.1-8B-Instruct and Mistral-7B-Instruct-v0.2. Notably, performance gaps widen as the cache budget decreases, highlighting the robustness of CentroidKV under aggressive compression. Detailed results in Tables 2 and 3 corroborate these observations: CentroidKV attains the best or near-best performance on most datasets and budgets, particularly on S-NIAH-1/2, MK-NIAH-1, MQ-NIAH, and VT. Performance degradation is observed on S-NIAH-3 and MK-NIAH-3, which involve UUID-type keys and values consisting of semantically arbitrary random strings. Because CentroidKV clusters tokens based on cosine similarity of key states, it is effective at preserving tokens with distinctive semantic representations (e.g., words or numbers), but struggles to differentiate tokens that lack meaningful semantic structure. As a result, UUID tokens are not reliably retained. This limitation is shared by other content-aware methods such as SnapKV and PyramidKV, whereas StreamingLLM remains stable by preserving tokens based on fixed positional rules rather than content.

**LongBench.** Figure 5 reports average results over 16 LongBench datasets. To provide a more fine-grained analysis, we further group datasets into six task categories (Appendix C) and report macro-averaged results in

Table 5: Task-average scores of LongBench across various budgets on Llama-3.1-8B-Instruct.

| Budget | Method | Single-Doc QA | Multi-Doc QA | Summarization | Few-shot | Synthetic | Code |
|---|---|---|---|---|---|---|---|
| 100% | Full | 44.90 | 48.21 | 29.17 | 53.83 | 55.35 | 50.27 |
| 75% | StreamingLLM | 36.18 | 42.01 | 27.53 | 53.87 | 42.27 | 47.13 |
| | SnapKV | 43.78 | 47.38 | 28.36 | 54.28 | 55.35 | **50.86** |
| | PyramidKV | 44.02 | 47.32 | 28.25 | 54.22 | 55.35 | 50.48 |
| | CentroidKV | **44.86** | **47.40** | **28.49** | **57.11** | **56.58** | 50.80 |
| 50% | StreamingLLM | 31.27 | 37.54 | 26.08 | 53.62 | 29.98 | 48.80 |
| | SnapKV | 38.73 | 45.20 | 26.96 | 56.27 | 55.27 | 50.28 |
| | PyramidKV | 39.63 | **45.38** | **27.18** | 54.52 | 54.75 | **51.31** |
| | CentroidKV | **42.71** | 43.66 | 27.04 | **57.57** | **55.75** | 50.86 |
| 25% | StreamingLLM | 24.42 | 29.98 | 24.46 | 53.11 | 20.25 | 50.67 |
| | SnapKV | 31.60 | **41.93** | **25.00** | **56.66** | 49.55 | **50.77** |
| | PyramidKV | 30.60 | 40.05 | 24.61 | 56.00 | **51.52** | 50.45 |
| | CentroidKV | **33.37** | 36.95 | 24.59 | 55.29 | 51.00 | 50.25 |

Table 6: Latency and memory comparison across varying context lengths on Llama-3.1-8B-Instruct.

| Context Length | Method | TTFT (s) | TPOT (ms) | Speedup ↑ | KV Cache (GB) |
|---|---|---|---|---|---|
| 32k | Full | 4.006 | 43.64 | 1× | 4.03 |
| | SnapKV | 4.096 | 35.21 | 1.24× | 1.10 |
| | StreamingLLM | 4.000 | 35.08 | 1.24× | 1.10 |
| | PyramidKV | 4.109 | 35.48 | 1.23× | 1.10 |
| | CentroidKV | 4.267 | **34.72** | **1.26×** | 1.10 |
| 64k | Full | 10.837 | 57.29 | 1× | 7.93 |
| | SnapKV | 11.057 | 37.10 | 1.54× | 2.08 |
| | StreamingLLM | 10.860 | **36.87** | **1.55×** | 2.08 |
| | PyramidKV | 11.057 | 38.11 | 1.50× | 2.08 |
| | CentroidKV | 11.291 | 37.87 | 1.51× | 2.08 |
| 128k | Full | 34.938 | 85.12 | 1× | 15.75 |
| | SnapKV | 34.551 | 44.32 | 1.92× | 4.03 |
| | StreamingLLM | 34.145 | **44.30** | **1.92×** | 4.03 |
| | PyramidKV | 34.532 | 44.42 | 1.92× | 4.03 |
| | CentroidKV | 35.001 | 44.57 | 1.91× | 4.03 |

Tables 4 and 5. CentroidKV performs consistently well on Single-Doc QA, Multi-Doc QA, and Summarization, achieving the best or near-best category-level averages under most budgets for both models. In contrast, performance drops are observed on Synthetic tasks and certain few-shot retrieval-style tasks, particularly on Mistral. Per-dataset results at the 25% budget (Tables 10 and 11) show that these degradations are concentrated in fine-grained retrieval and index-sensitive tasks. In such settings, aggressive token merging can obscure exact token matches and positional signals that are critical for retrieval-based queries. By contrast, CentroidKV remains highly effective on tasks requiring semantic aggregation and reasoning, where preserving high-level contextual information is more important than exact token fidelity.

**Summary.** Overall, CentroidKV delivers strong accuracy across both benchmarks, particularly under constrained KV cache budgets. Its strengths are most evident in tasks requiring semantic understanding, while its limitations arise in retrieval-intensive scenarios that depend on precise token-level information.

Table 7: Latency and memory comparison across varying context lengths on Mistral-7B-Instruct.

| Context Length | Method | TTFT (s) | TPOT (ms) | Speedup ↑ | KV Cache (GB) |
|---|---|---|---|---|---|
| 32k | Full | 3.920 | 43.42 | 1× | 4.03 |
| | SnapKV | 4.027 | **34.23** | **1.27×** | 1.10 |
| | StreamingLLM | 3.918 | 34.95 | 1.24× | 1.10 |
| | PyramidKV | 3.961 | 35.31 | 1.23× | 1.10 |
| | CentroidKV | 4.184 | 34.79 | 1.25× | 1.10 |
| 64k | Full | 10.650 | 57.09 | 1× | 7.93 |
| | SnapKV | 10.872 | 37.12 | 1.54× | 2.08 |
| | StreamingLLM | 10.637 | **36.98** | **1.54×** | 2.08 |
| | PyramidKV | 10.884 | 37.49 | 1.52× | 2.08 |
| | CentroidKV | 11.111 | 37.23 | 1.53× | 2.08 |
| 128k | Full | 34.661 | 84.71 | 1× | 15.75 |
| | SnapKV | 33.897 | 44.19 | 1.92× | 4.03 |
| | StreamingLLM | 33.802 | 44.54 | 1.90× | 4.03 |
| | PyramidKV | 34.182 | 44.50 | 1.90× | 4.03 |
| | CentroidKV | 34.668 | **44.12** | **1.92×** | 4.03 |

Table 8: Ablation study (RULER score and TTFT(s)) on chunking strategy and varied chunk size.

| Chunk Size | Avg. Score | 16k | 32k | 64k | 128k |
|---|---|---|---|---|---|
| 64 | 59.76 | 1.791 | 4.195 | 11.316 | 34.849 |
| 128 | 59.06 | 1.792 | 4.268 | 11.284 | 34.856 |
| 256 | 59.87 | 1.815 | 4.257 | 11.301 | 34.896 |
| w/o chunk | 60.62 | 1.951 | 4.941 | 14.415 | OOM |

### 6.3 Efficiency Evaluation

We evaluate inference efficiency in terms of decoding latency and GPU memory footprint. Following standard practice, we report Time To First Token (TTFT) to capture pre-filling overhead, and Time Per Output Token (TPOT) to measure decoding efficiency. All compression-based methods are configured with a uniform 25% KV cache budget and evaluated against the full-attention baseline under an identical inference pipeline. In addition, we integrate CentroidKV into the vLLM serving framework and report the results in Appendix D, where CentroidKV improves serving throughput by up to 4.0×.

**Latency.** Tables 6 and 7 present latency across context lengths ranging from 32K to 128K, with a fixed decoding length of 1K tokens. Overall, CentroidKV achieves competitive decoding efficiency while incurring only modest pre-filling overhead. Specifically, TTFT is slightly higher than full baseline and prior compression methods due to the additional online clustering operations. Nevertheless, the overhead remains consistently small across all evaluated settings. During decoding, CentroidKV achieves speedups comparable to prior compression methods, achieving up to 1.91× TPOT reduction on Llama and 1.92× on Mistral at 128K context length. This is as expected since all methods operate under the same effective KV cache budget.

**Memory.** As shown in Tables 6 and 7, all compression-based methods, including CentroidKV, reduce KV cache memory by approximately 4× compared to full attention. These results demonstrate that CentroidKV achieves memory efficiency comparable to prior methods.

**Summary.** Overall, CentroidKV achieves competitive latency and memory efficiency compared to prior compression-based methods, while consistently delivering improved accuracy under the same KV cache budget.

Table 9: Ablation study (RULER score and TTFT(s)) on scheduling of compression ratio defined in Eq. 4.

| $r_{\mathrm{init}}$ | $\delta_r$ | Avg. Score | 16k | 32k | 64k | 128k |
|---|---|---|---|---|---|---|
| | 0.00 | 19.22 | 1.715 | 4.110 | 11.105 | 34.601 |
| 1.00 | 0.10 | 23.77 | 1.720 | 4.113 | 11.106 | 34.650 |
| | 0.20 | 25.80 | 1.718 | 4.122 | 11.092 | 34.674 |
| | 0.30 | 26.69 | 1.754 | 4.142 | 11.129 | 34.717 |
| | 0.00 | 47.29 | 1.717 | 4.114 | 11.077 | 34.648 |
| 0.90 | 0.10 | 56.25 | 1.720 | 4.118 | 11.098 | 34.606 |
| | 0.20 | 59.26 | 1.734 | 4.139 | 11.157 | 34.779 |
| | 0.30 | 56.77 | 1.761 | 4.170 | 11.223 | 34.777 |
| | 0.00 | 51.44 | 1.723 | 4.186 | 11.088 | 34.688 |
| 0.80 | 0.10 | 58.97 | 1.738 | 4.116 | 11.152 | 34.710 |
| | 0.20 | 60.46 | 1.786 | 4.168 | 11.274 | 34.910 |
| | 0.30 | 57.33 | 1.840 | 4.264 | 11.364 | 35.100 |

## 6.4 Ablation Study

We conduct ablations on both accuracy and efficiency to analyze the key design choices in CentroidKV, focusing on chunked clustering strategy and the scheduling of compression ratio. Results are reported using average accuracy on RULER and TTFT across varying context lengths.

**Chunking.** Table 8 evaluates the impact of chunk size $c$ in Algorithm 1. Without chunking, clustering is performed over the full sequence, leading to the highest accuracy but incurs substantial overhead and fails to scale to long contexts (OOM at 128K). Introducing chunking reduces the matching scope from the entire sequence to local segments, significantly lowering latency and improving scalability. Across chunked variants, $c = 256$ achieves the best trade-off, attaining the highest average score while maintaining comparable TTFT. Smaller chunks slightly degrade accuracy, likely due to restricted matching scope within each chunk. These results demonstrate that chunked clustering effectively amortizes computational cost with minimal impact on accuracy, and that a moderate chunk size provides the best balance between efficiency and representational capacity.

**Compression Ratio.** Table 9 evaluates the compression ratio scheduling defined by Equation 4. We observe that aggressive merging with $r_{\mathrm{init}} = 1.0$ leads to severe accuracy degradation, indicating that indiscriminate merging introduces substantial noise by aggregating low-similarity tokens. Reducing $r_{\mathrm{init}}$ to 0.9 or 0.8 significantly improves performance, as it prioritizes higher-confidence matches. Applying a decaying schedule further enhances accuracy. As clustering proceeds, decreasing $r$ progressively enforces stricter merging criteria, effectively filtering out weaker similarities in later rounds. This leads to more reliable centroid representations across iterations. The best configuration ($r_{\mathrm{init}} = 0.80$, $\delta_r = 0.20$) achieves the highest average score while maintaining comparable TTFT across all context lengths.

## 7 Conclusion and Limitations

This paper presents CentroidKV, a simple yet effective framework for online KV cache clustering that improves the efficiency of long-context LLM inference. CentroidKV reduces KV cache memory usage by up to 75% without compromising accuracy on most tasks, while accelerating the decoding stage by up to 1.92× and increasing the throughput by up to 4×. However, our work focuses on compressing the KV cache on the GPU and does not investigate memory offloading strategies. A promising direction for future research is to perform clustering on the CPU and transfer the resulting centroids to the GPU. Additionally, CentroidKV currently employs a manually specified compression-ratio schedule and restricts cache reduction to at most half at each clustering step. This could be addressed by designing an adaptive compression strategy in future work.

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

## A  Theoretical Results and Proof

We theoretically justify the optimality of our partitioning strategy which divides each chunk into two sets $A$ and $B$ in an alternating manner. Intuitively, the partitioning strategy should retain the edges with the highest similarities to yield better clusters. Based on Observation 2 in Section 4, we consider a convex and monotonically decreasing score function $f : \mathbb{N} \to \mathbb{R}$ that maps the distance to an importance score. With the score function $f$, the optimal partitioning can be achieved by solving the following optimization problem:

$$\max_{A,B} \quad \sum_{x \in A} \sum_{y \in B} f(|x - y|). \tag{6}$$

The following theorem states that, for any score function $f$ that is convex and monotonically decreasing, the alternating partitioning strategy we propose is always the optimal solution of problem Equation (6).

**Theorem A.1.** *Define partition set* $\mathcal{P}_{2n} = \{(A, B) \mid |A| = |B| = n, \text{ and } A \cup B = [2n]\}$. *If function* $f : [2n-1] \to \mathbb{R}$ *satisfies* $f(1) - f(2) \geq f(2) - f(3) \geq \cdots \geq f(2n-2) - f(2n-1) \geq 0$, *it holds that*

$$(A_0, B_0) = (\{1, 3, \cdots, 2n-1\}, \{2, 4, \cdots, 2n\})$$

$$\in \underset{(A,B) \in \mathcal{P}_{2n}}{\arg\max} \sum_{x \in A} \sum_{y \in B} f(|x - y|).$$

*Here, we use notation* $[k]$ *to denote the set of positive integers no larger than* $k$, *i.e.,* $[k] := [1, k] \cap \mathbb{Z}$.

*Proof.* When $n = 1$, the result is trivial. In the following, we assume $n \geq 2$. Consider the following mapping:

$$\phi_{2n} : \mathbb{Z} \to [2n]$$

$$x \mapsto \begin{cases} 1, & x < 1; \\ x, & x \in [2n]; \\ 2n, & x > 2n; \end{cases}$$

For any $l \in [2n-1]$, define $\mathcal{S}_{2n,l} := \{(\phi_{2n}(x), \phi_{2n}(x+l)) \mid x \in \mathbb{Z} \cap [-l+2, 2n-1]\}$. For $\forall (A, B) \in \mathcal{P}_{2n}$, define

$$\mathcal{E}_{A,B} := \{(\min\{a, b\}, \max\{a, b\}) \mid a \in A \text{ and } b \in B\}, \tag{7}$$

$$\mathcal{D}_{A,B,l} := \{(a, b) \in \mathcal{E}_{A,B} \mid b - a = l\}, \tag{8}$$

$$\mathcal{T}_{A,B,l} := \{(a, b; c, d) \in [2n]^4 \mid (a, b) \in \mathcal{S}_{2n,l}, \ (c, d) \in \mathcal{E}_{A,B}, \text{ and } [c, d] \subseteq [a, b]\}, \tag{9}$$

Now we calculate the number of elements in $\mathcal{T}_{A,B,l}$, *i.e.*, $|\mathcal{T}_{A,B,l}|$. Define $\mathcal{K}_{A,B,a,b} := \{(c,d) \in \mathcal{E}_{A,B} \mid [c,d] \subseteq [a,b]\}$, it holds that

$$|\mathcal{T}_{A,B,l}| = \sum_{(a,b) \in \mathcal{S}_{2n,l}} |\mathcal{K}_{A,B,a,b}|. \tag{10}$$

Note that

$$|\mathcal{K}_{A,B,a,b}| = |[a,b] \cap A| \cdot |[a,b] \cap B| \leq \left\lfloor \frac{b-a+1}{2} \right\rfloor \cdot \left\lceil \frac{b-a+1}{2} \right\rceil, \tag{11}$$

applying equation 11 to equation 10 yields

$$|\mathcal{T}_{A,B,l}| \leq \sum_{(a,b) \in \mathcal{S}_{2n,l}} \left\lfloor \frac{b-a+1}{2} \right\rfloor \cdot \left\lceil \frac{b-a+1}{2} \right\rceil$$

$$= \sum_{i=2}^{l} \left\lfloor \frac{i-1+1}{2} \right\rfloor \cdot \left\lceil \frac{i-1+1}{2} \right\rceil + \sum_{i=1}^{2n-l} \left\lfloor \frac{(i+l)-i+1}{2} \right\rfloor \cdot \left\lceil \frac{(i+l)-i+1}{2} \right\rceil +$$

$$+ \sum_{i=2n-l+1}^{2n-1} \left\lfloor \frac{2n-i+1}{2} \right\rfloor \cdot \left\lceil \frac{2n-i+1}{2} \right\rceil$$

$$= \begin{cases} -\frac{1}{12}l^3 + \frac{2n-1}{4}l^2 + \frac{6n-1}{6}l, & 2 \mid l; \\ -\frac{1}{12}l^3 + \frac{2n-1}{4}l^2 + \frac{12n-5}{12}l + \frac{2n-1}{4}, & 2 \nmid l. \end{cases} \triangleq c_{2n,l}. \tag{12}$$

On the other hand, define $\mathcal{K}'_{2n,l,c,d} := \{(a,b) \in \mathcal{S}_{2n,l} \mid [c,d] \subseteq [a,b]\}$, we have

$$|\mathcal{K}'_{2n,l,c,d}| = \max(l+1-d+c, 0),$$

thus

$$|\mathcal{T}_{A,B,l}| = \sum_{(c,d) \in \mathcal{E}_{A,B}} |\mathcal{K}'_{2n,l,c,d}|$$

$$= \sum_{(c,d) \in \mathcal{E}_{A,B}} \max(l+1-d+c, 0)$$

$$= \sum_{i=1}^{l} (l+1-i)|\mathcal{D}_{A,B,i}|. \tag{13}$$

Combining equation 12equation 13 yields

$$\sum_{i=1}^{l} (l+1-i)|\mathcal{D}_{A,B,i}| \leq c_{2n,l}. \tag{14}$$

Define $a_i = f(i) - f(i+1)$ for $i \in [2n-2]$, $b_i = a_i - a_{i+1}$ for $i \in [2n-3]$, and let $b_{2n-2} = a_{2n-2}$, it holds that $b_1, b_2, \cdots, b_{2n-2} \geq 0$. Note that

$$f(i) = f(2n-1) + \sum_{j=i}^{2n-2} a_j,$$

$$= f(2n-1) + \sum_{j=i}^{2n-2} \sum_{k=j}^{2n-2} b_k,$$

$$= f(2n-1) + \sum_{j=i}^{2n-2} (j-i+1)b_j, \quad \forall i \in [2n-2],$$

we have

$$\sum_{x \in A} \sum_{y \in B} f(|x - y|) = \sum_{i=1}^{2n-1} |\mathcal{D}_{A,B,i}| f(i)$$

$$= \sum_{i=1}^{2n-1} |\mathcal{D}_{A,B,i}| \left( f(2n-1) + \sum_{j=i}^{2n-2} (j - i + 1) b_j \right)$$

$$= |\mathcal{E}_{A,B}| f(2n-1) + \sum_{j=1}^{2n-2} \sum_{i=1}^{j} (j + 1 - i) |\mathcal{D}_{A,B,i}| b_j$$

$$\leq n^2 f(2n-1) + \sum_{j=1}^{2n-2} c_{2n,j} b_j, \tag{15}$$

where the last inequality uses $|\mathcal{E}_{A,B}| = n^2$ and equation 14. If $A_0 = \{1, 3, \cdots, 2n-1\}$ and $B_0 = \{2, 4, \cdots, 2n\}$, we have

$$\sum_{x \in A_0} \sum_{y \in B_0} f(|x - y|) = \sum_{i=1}^{2n-1} |\mathcal{D}_{A_0,B_0,i}| f(i) = \sum_{i=1}^{n} |\mathcal{D}_{A_0,B_0,2i-1}| f(2i-1)$$

$$= \sum_{i=1}^{n} (2n - 2i + 1) \left( f(2n-1) + \sum_{j=2i-1}^{2n-2} (j - 2i + 2) b_j \right)$$

$$= n^2 f(2n-1) + \sum_{j=1}^{2n-2} \sum_{i=1}^{\lfloor \frac{j+1}{2} \rfloor} (2n - 2i + 1)(j - 2i + 2) b_j. \tag{16}$$

Note that

$$\sum_{i=1}^{\lfloor \frac{j+1}{2} \rfloor} (2n - 2i + 1)(j - 2i + 2) = \begin{cases} -\frac{1}{12} l^3 + \frac{2n-1}{4} l^2 + \frac{6n-1}{6} l, & 2 \mid l; \\ -\frac{1}{12} l^3 + \frac{2n-1}{4} l^2 + \frac{12n-5}{12} l + \frac{2n-1}{4}, & 2 \nmid l. \end{cases} = c_{2n,j}, \tag{17}$$

combining equation 16equation 17 yields

$$\sum_{x \in A_0} \sum_{y \in B_0} f(|x - y|) = n^2 f(2n-1) + \sum_{j=1}^{2n-2} c_{2n,j} b_j. \tag{18}$$

Combining equation 15equation 18, we obtain

$$(A_0, B_0) \in \operatorname*{arg\,max}_{(A,B) \in \mathcal{P}_{2n}} \sum_{x \in A} \sum_{y \in B} f(|x - y|),$$

which concludes the proof. $\qquad \square$

## B  Experimental Setups

We compare CentroidKV with other compression-based baselines based on KVPress Devoto et al. (2025), which implements multiple KV cache compression methods and benchmarks. All baseline methods are evaluated using their recommended settings. For StreamingLLM (Xiao et al., 2023), the number of initial sink tokens is 4. For SnapKV (Li et al., 2024), window size is 64 and pooling kernel size is 5. For PyramidKV (Cai et al., 2024), the hyperparameter which controls the pyramid's shape and steepness is set to 20.

Parameters of CentroidKV maintain the same across the entire evaluation process, which are reported in the implementation details of Section 6. Ablation studies on chunk size and the compression ratio (including $r_{\text{init}}$ and $\delta_r$) are demonstrated in Section 6. For number of attention sink and recent tokens, we simply follow the recommended settings in DuoAttention Xiao et al. (2024).

Table 10: Detailed results of LongBench datasets with 25% KV cache budget on Llama-3.1-8B-Instruct.

| Dataset | StreamingLLM | SnapKV | PyramidKV | CentroidKV | Full |
|---|---|---|---|---|---|
| 2WikiMQA | 27.92 | **42.44** | 39.83 | 33.18 | 51.33 |
| GovReport | 28.79 | **29.24** | 28.23 | 27.24 | 35.19 |
| HotpotQA | 41.95 | **55.18** | 55.13 | 52.08 | 59.80 |
| LCC | 50.84 | **53.83** | 53.61 | 51.21 | 53.31 |
| MultiNews | **23.91** | 23.35 | 23.41 | 23.10 | 26.99 |
| MultiFieldQA-en | 25.62 | 35.01 | 34.79 | **39.81** | 56.31 |
| Musique | 20.07 | **28.18** | 25.19 | 25.58 | 33.51 |
| NarrativeQA | 23.25 | **28.43** | 26.92 | 27.03 | 30.99 |
| Passage Count | 7.00 | 9.10 | 9.55 | **10.00** | 10.70 |
| PassageRetrieval-en | 33.50 | 90.00 | **93.50** | 92.00 | 100.00 |
| Qasper | 24.38 | 31.36 | 30.08 | **33.27** | 47.39 |
| QMSum | 20.69 | 22.41 | 22.20 | **23.42** | 25.34 |
| RepoBench-P | **50.50** | 47.72 | 47.28 | 49.30 | 47.23 |
| SAMSum | 35.70 | 41.43 | **42.26** | 41.74 | 40.77 |
| TREC | 31.50 | **37.00** | 34.00 | 33.00 | 29.00 |
| TriviaQA | **92.12** | 91.54 | 91.73 | 91.14 | 91.71 |

Table 11: Detailed results of LongBench datasets with 25% KV cache budget on Mistral-7B-Instruct-v0.2.

| Dataset | StreamingLLM | SnapKV | PyramidKV | CentroidKV | Full |
|---|---|---|---|---|---|
| 2WikiMQA | **15.83** | 13.75 | 13.59 | 15.62 | 20.97 |
| GovReport | **28.38** | 27.27 | 26.20 | 26.87 | 32.34 |
| HotpotQA | 21.59 | 23.96 | 22.98 | **25.98** | 35.34 |
| LCC | 48.37 | **51.25** | 50.79 | 47.34 | 51.35 |
| MultiNews | 22.87 | 23.16 | 23.23 | **23.60** | 26.55 |
| MultiFieldQA-en | 22.46 | 31.65 | 31.27 | **32.20** | 45.94 |
| Musique | 10.42 | **11.32** | 10.03 | 9.68 | 17.30 |
| NarrativeQA | 21.36 | 18.62 | 17.30 | **21.42** | 23.86 |
| Passage Count | 3.10 | **3.91** | 3.39 | 2.94 | 2.48 |
| PassageRetrieval-en | 18.48 | **68.81** | 66.77 | 35.32 | 73.76 |
| Qasper | 13.32 | 14.36 | 13.57 | **15.45** | 29.20 |
| QMSum | 20.58 | 21.59 | 21.41 | **22.94** | 24.37 |
| RepoBench-P | 47.55 | **51.08** | 50.74 | 50.09 | 51.16 |
| SAMSum | 36.12 | 38.68 | **39.10** | 36.63 | 39.61 |
| TREC | **47.50** | 41.75 | 39.25 | 23.50 | 51.25 |
| TriviaQA | 48.69 | **79.43** | 77.46 | 67.37 | 74.68 |

## C   Additional Results of LongBench

Below is the overview of datasets categorized by tasks in LongBench benchmark.

- Single-document QA: NarrativeQA, Qasper, MultiFieldQA-en.

- Multi-document QA: HotpotQA, 2WikiMQA, Musique.

- Summarization: GovReport, QMSum, MultiNews.

- Few-shot learning: SAMSum, TREC, TriviaQA.

- Synthetic Tasks: PassageRetrieval-en, Passage Count.

- Code Completion: LCC, RepoBench-P.

Table 12: Efficiency evaluation within vLLM on Llama-3.1-8B-Instruct, 25% KV cache budget of CentroidKV.

| Context | TPOT (ms) | | | Throughput (tok/s) | | |
|---------|------|-----------|---------|------|-----------|-------|
| Length | Full | CentroidKV | Speedup | Full | CentroidKV | Gain |
| 4k | 13.91 | 13.65 | 1.02× | 1263.8 | 2960.2 | 2.34× |
| 8k | 14.27 | 13.59 | 1.05× | 669.3 | 2255.2 | 3.37× |
| 16k | 15.05 | 13.81 | 1.09× | 388.1 | 1335.6 | 3.44× |
| 32k | 16.73 | 14.47 | 1.16× | 203.7 | 664.7 | 3.26× |
| 64k | 20.47 | 14.93 | 1.37× | 100.5 | 401.7 | 4.00× |
| 128k | 26.79 | 16.43 | 1.63× | 50.4 | 155.2 | 3.08× |

Table 10 and 11 present the detailed results of different methods across 16 LongBench datasets on Llama and Mistral respectively. CentroidKV outperforms on semantic aggregation and reasoning-heavy tasks, while showing limitations on retrieval-intensive tasks requiring precise token preservation.

## D   Efficiency evaluation based on vLLM

To validate that CentroidKV's compression benefits persist under realistic serving conditions with memory management overhead, we conduct experiments within vLLM Kwon et al. (2023), a widely deployed production inference engine that employs PagedAttention for dynamic KV cache memory management.

We integrate CentroidKV into vLLM's inference pipeline as a post-prefill compression step: after the prefill phase populates the paged KV cache, CentroidKV reads the key-value tensors, applies compression and writes the compressed representations back into the page table. Freed memory blocks are returned to the block manager, enabling the engine to serve additional concurrent requests. All experiments use Llama-3.1-8B-Instruct on a single A100 GPU. We evaluate across context lengths from 4K to 128K tokens and report two metrics: (1) Time Per Output Token (TPOT) at batch size 1, measuring single-request decode latency; (2) Peak generation throughput, measuring the maximum number of concurrent requests that fit within GPU memory.

As shown in Table 12, CentroidKV achieves decoding speedups ranging from 1.02× at 4K context to 1.63× at 128K context. This confirms that the reduced KV cache footprint directly translates into lower memory-bandwidth pressure during attention computation, with the benefit growing as context length increases and attention becomes increasingly memory-bound. The throughput improvements are substantially larger. By freeing 75% of KV cache blocks after compression, CentroidKV enables the engine to sustain 2.3–4.0× more concurrent requests within the same GPU memory.

## E   Comparison with ClusterKV

ClusterKV (Liu et al., 2024a) and CentroidKV both use clustering but for undamentally different design objectives. ClusterKV uses clustering as an recall mechanism while retaining the full KV cache, whereas CentroidKV directly replaces raw KV states with cluster centroids as a compressed representation. As a result, ClusterKV prioritizes information preservation and can be viewed as an accuracy-oriented retrieval baseline, while CentroidKV explicitly targets KV memory reduction and decoding efficiency.

To make this distinction explicit, we evaluate ClusterKV using its official implementation (with custom CUDA kernels) and compare both methods against their respective dense-KV baselines on Llama-3.1-8B-Instruct. Results are reported in Table 13 and Table 14. All experiments use FP16 precision, a fixed KV cache budget of 2048, and a decoding length of 256 across varying context lengths. We exclude the 64k setting for ClusterKV due to infeasibility.

Since the two methods rely on different system implementations, we focus on their relative overhead with respect to their own full-KV baselines. Under this protocol, ClusterKV consistently introduces additional

Table 13: CentroidKV vs. full KV on Llama-3.1-8B-Instruct with uniform KV cache budget 2048, evaluated in KVPress codebase.

| Context Length | Method | TTFT (s) | TPOT (ms) | Speedup | Memory (GB) | Mem Saving |
|:---:|:---|:---:|:---:|:---:|:---:|:---:|
| 4k | Full | 0.371 | 35.53 | $\times$ | 16.44 | $\checkmark$ |
| | CentroidKV | 0.444 | 39.23 | | 16.20 | |
| 8k | Full | 0.780 | 35.82 | $\times$ | 17.92 | $\checkmark$ |
| | CentroidKV | 0.963 | 37.80 | | 17.19 | |
| 16k | Full | 1.735 | 38.47 | $\checkmark$ | 20.87 | $\checkmark$ |
| | CentroidKV | 2.036 | **36.58** | | 19.24 | |
| 32k | Full | 4.186 | 44.67 | $\checkmark$ | 26.76 | $\checkmark$ |
| | CentroidKV | 4.513 | **35.34** | | 23.49 | |
| 64k | Full | 11.300 | 59.02 | $\checkmark$ | 38.56 | $\checkmark$ |
| | CentroidKV | 11.881 | **35.44** | | 32.01 | |

Table 14: ClusterKV vs. full KV on Llama-3.1-8B-Instruct with uniform KV cache budget 2048, evaluated in ClusterKV's official codebase with their customized kernels.

| Context Length | Method | TTFT (s) | TPOT (ms) | Speedup | Memory (GB) | Mem Saving |
|:---:|:---|:---:|:---:|:---:|:---:|:---:|
| 4k | Full | 0.355 | 13.89 | $\times$ | 18.46 | $\times$ |
| | ClusterKV | 0.453 | 16.44 | | 18.50 | |
| 8k | Full | 0.758 | 14.01 | $\times$ | 21.84 | $\times$ |
| | ClusterKV | 0.962 | 16.45 | | 21.89 | |
| 16k | Full | 1.683 | 14.92 | $\times$ | 28.63 | $\times$ |
| | ClusterKV | 2.084 | 16.65 | | 28.69 | |
| 32k | Full | 4.179 | 16.30 | $\times$ | 42.16 | $\times$ |
| | ClusterKV | 5.134 | 17.05 | | 42.25 | |

overhead without improving memory efficiency. As shown in Table 14, TTFT increases by 22.9%–27.4% across 4k to 32k contexts, while TPOT degrades by 4.6%–18.4%. Moreover, end-to-end GPU memory usage remains effectively unchanged (or slightly higher), indicating that ClusterKV does not achieve actual KV footprint reduction.

In contrast, CentroidKV exhibits a different efficiency profile. As shown in Table 13, it incurs a moderate increase in TTFT due to the compression cost, but this overhead does not scale with context length. More importantly, CentroidKV significantly improves decoding efficiency at long contexts: TPOT becomes lower than full KV starting from 16k, yielding substantial per-token speedups. In addition, it achieves consistent memory savings that grow with context length, reducing GPU memory by 1.5% at 4k and up to 17.0% at 64k.

Overall, while both methods exploit similarity in the key space, their system-level trade-offs differ substantially. CentroidKV translates redundancy into compact in-GPU representations, leading to tangible gains in both memory and long-context decoding efficiency. In contrast, ClusterKV maintains a recallable clustered index on top of the full KV cache, where the additional indexing, selection, and data movement introduce persistent overhead without reducing the underlying memory footprint.

# F   Statistical Significance Analysis

Given that aggregate scores across benchmarks sometimes differ by only a few percentage points, we conduct bootstrap hypothesis testing to verify that the observed performance differences are statistically meaningful rather than artifacts of evaluation variance. We employ the non-parametric bootstrap (Efron & Tibshirani,

Table 15: Bootstrap summary on RULER for Llama-3.1-8B-Instruct. Values are reported as mean $\pm$ std.

| Method | 75% | 50% | 25% |
|---|---|---|---|
| StreamingLLM | $80.69 \pm 0.96$ | $58.59 \pm 1.16$ | $37.20 \pm 1.02$ |
| SnapKV | $80.97 \pm 0.80$ | $68.67 \pm 0.90$ | $41.78 \pm 0.94$ |
| PyramidKV | $81.95 \pm 0.77$ | $68.33 \pm 0.90$ | $41.59 \pm 0.93$ |
| CentroidKV | $\mathbf{90.66 \pm 0.70}$ | $\mathbf{79.24 \pm 0.77}$ | $\mathbf{60.26 \pm 0.78}$ |

Table 16: Bootstrap summary on RULER for Mistral-7B-Instruct-v0.2. Values are reported as mean $\pm$ std.

| Method | 75% | 50% | 25% |
|---|---|---|---|
| StreamingLLM | $75.42 \pm 1.01$ | $56.57 \pm 1.13$ | $37.64 \pm 1.04$ |
| SnapKV | $58.38 \pm 1.01$ | $42.06 \pm 0.99$ | $32.77 \pm 0.89$ |
| PyramidKV | $62.21 \pm 1.01$ | $39.81 \pm 0.95$ | $30.58 \pm 0.90$ |
| CentroidKV | $\mathbf{82.60 \pm 0.84}$ | $\mathbf{58.81 \pm 0.99}$ | $\mathbf{38.02 \pm 0.78}$ |

1994) to estimate the sampling distribution of aggregate scores. We report bootstrap standard deviations and 95% percentile confidence intervals on RULER and LongBench on both models.

On RULER for Llama-3.1-8B-Instruct (Table 15), at the most aggressive compression (25% budget), CentroidKV achieves $60.26 \pm 0.78$, with a 95% CI of $[58.73, 61.80]$, while the next-best methods PyramidKV ($41.59 \pm 0.93$, CI $[39.74, 43.42]$) and SnapKV ($41.78 \pm 0.94$, CI $[39.92, 43.62]$). The confidence intervals are entirely non-overlapping, confirming that the advantage is statistically significant. This separation is consistent at different compression ratios and models shown in Table 16.

On LongBench with Llama-3.1-8B-Instruct at 25% budget (Table 17), CentroidKV performs comparably to PyramidKV and SnapKV, with overlapping CIs indicating no statistically significant difference among the top three. On Mistral (Table 18), a similar pattern emerges at moderate cache budget. While for aggressive compression, as analyzed in the main body of paper, the performance degradation of CentroidKV comes from several retrieval-intensive tasks.

Table 17: Bootstrap summary on LongBench for Llama-3.1-8B-Instruct. Values are reported as mean $\pm$ std.

| Method | 75% | 50% | 25% |
|---|---|---|---|
| StreamingLLM | $41.10 \pm 0.56$ | $37.69 \pm 0.56$ | $33.61 \pm 0.52$ |
| SnapKV | $45.86 \pm 0.53$ | $44.54 \pm 0.53$ | $\mathbf{41.64 \pm 0.54}$ |
| PyramidKV | $45.82 \pm 0.53$ | $44.52 \pm 0.52$ | $41.11 \pm 0.52$ |
| CentroidKV | $\mathbf{46.77 \pm 0.54}$ | $\mathbf{45.39 \pm 0.53}$ | $40.82 \pm 0.53$ |

Table 18: Bootstrap summary on LongBench for Mistral-7B-Instruct-v0.2. Values are reported as mean $\pm$ std.

| Method | 75% | 50% | 25% |
|---|---|---|---|
| StreamingLLM | $33.50 \pm 0.49$ | $30.30 \pm 0.48$ | $26.66 \pm 0.44$ |
| SnapKV | $\mathbf{36.98 \pm 0.47}$ | $\mathbf{35.64 \pm 0.46}$ | $\mathbf{32.55 \pm 0.45}$ |
| PyramidKV | $36.65 \pm 0.48$ | $33.99 \pm 0.45$ | $31.70 \pm 0.44$ |
| CentroidKV | $36.42 \pm 0.49$ | $34.35 \pm 0.48$ | $28.56 \pm 0.45$ |

