# OpenReview forum: "CentroidKV: Efficient Long-Context LLM Inference via KV Cache Clustering"
_TMLR — Accepted by TMLR_

### Review · Reviewer_6a5h · 2026-03-31

**Summary Of Contributions:**

CentroidKV compresses transformer KV caches by dividing the sequence into fixed-size chunks, applying an alternating (odd/even) split with chunked soft matching to cluster tokens, and storing only the centroid per cluster. The paper argues this yields substantial memory savings and throughput gains, and it supplies a combinatorial proof that the alternating split is optimal for their within-chunk objective. Evaluations on RULER, LongBench, and needle-in-a-haystack setups compare against StreamingLLM, SnapKV, and PyramidKV.

Key strengths:
- Strong performance on RULER, especially the VT task, and clean presentation of memory savings.
- Chunked compression pipeline with an accessible complexity analysis comparing against k-means and bipartite soft matching.
- The alternating-partition proof is correct and clarifies why the odd/even split maximizes their convex distance objective inside a chunk (even if the overall algorithm remains heuristic).

Key weaknesses:
- “Chunked soft matching” is a direct application of the alternating odd/even partition from bipartite soft matching, and prior KV compression papers already employ clustering (e.g., ClusterAttn, STARC, IceCache).
- The paper reports TTFT/TPOT improvements only against the full KV cache; there are no end-to-end latency or throughput numbers versus SnapKV or PyramidKV, despite reproducing those baselines for accuracy. That gap leaves the efficiency claims unsubstantiated.
- CentroidKV lags SnapKV substantially on several QA/retrieval tasks (passage retrieval, TREC, TriviaQA). Labeling these as “outliers” is unconvincing as there are many LongBench tasks that share their type.
- The paper doesn't explicitly state how much tuning went into each method's hyper-parameters (e.g. chunk size, thresholds, temperature). Without those ablations, it’s hard to assess robustness or reproducibility.

**Audience:**

Yes

**Audience Explanation:**

Indeed, long-context inference is a pressing issue for practitioners, and this paper provides concrete data points on memory–latency trade-offs for chunkwise KV compression. Even if the present evidence is incomplete, documenting where CentroidKV excels and where it struggles is valuable. With tighter claims and fuller comparisons, the work could serve as a useful reference for the community.

**Broader Impact Concerns:**

None.

**Claims And Evidence:**

No

**Claims Explanation:**

The empirical evidence does not fully substantiate the key claims:
- While CentroidKV matches full-cache latency in the reported TTFT/TPOT curves, the paper never measures end-to-end latency against competing compression methods like SnapKV or PyramidKV, so the claimed efficiency edge remains unchecked.
- On LongBench, CentroidKV trails SnapKV substantially on several QA and retrieval tasks (e.g., passage retrieval, TREC, TriviaQA), yet these regressions are dismissed as “outliers” without justification.
- Finally, the experimental section does not clarify how much tuning effort was invested in CentroidKV versus the baselines, leaving doubts about the comparisons.
Without stronger evidence, the assertion of broad superiority isn’t convincing enough for acceptance.

**Requested Changes:**

Critical:
- Add end-to-end latency measurements (TTFT, TPOT, throughput) comparing CentroidKV with SnapKV and PyramidKV under identical hardware and implementation assumptions; the current comparison to only the full cache leaves the efficiency claims unsubstantiated.
- Report macro averages/per-task deltas, and analyze the large gaps on certain QA/retrieval tasks. If the method is tuned for certain categories, document that and adjust claims accordingly.

Other:
- Document the tuning setup for CentroidKV and each baseline (hyperparameters, calibration data, number of trials) so readers can judge comparison fairness.
- Provide ablations for key hyper-parameters (chunk size, centroid count) to show how quality and latency respond; this would clarify robustness.
- Prior work already applies clustering to KV caches (e.g., ClusterAttn, STARC, IceCache). Position CentroidKV as a particular chunk-based adaptation rather than the first clustering method.
- Tighten the theory–practice link by explicitly stating in the main text how the alternating-partition proof maps onto the implemented chunked soft matching, or move the result to the appendix if it remains mostly illustrative.

---

> ### Author Response · Authors · 2026-05-07
> **Response to Reviewer 6a5h**
>
> We sincerely appreciate Reviewer 6a5h for the valuable comments. The concerns are addressed as follows.
>
> ---
> **1. End-to-end latency comparisons with SnapKV and PyramidKV**
>
> We have added comprehensive end-to-end latency comparisons in the revised **Tables 6 and 7**. These tables report TTFT, TPOT, speedup, and KV cache memory usage for all methods (Full, StreamingLLM, SnapKV, PyramidKV, and CentroidKV) under identical hardware and implementation settings (KVPress codebase, same GPU).
>
> The results show that at a fixed KV cache budget (e.g., 25%), all compression-based methods achieve comparable TPOT and decoding speedup, as the decoding stage is primarily memory-bound. CentroidKV consistently achieves higher accuracy while maintaining competitive latency, with only a modest TTFT overhead due to clustering. Accordingly, we have revised our claims to reflect that the primary advantage of CentroidKV lies in improving the accuracy–efficiency trade-off under the same compression budget, rather than reducing latency beyond other compression methods.
>
> ---
> **2. Macro averages and analysis of QA/retrieval gaps**
>
> We have expanded the evaluation to include macro-averaged results by task category in **Tables 4 and 5**, covering single-document QA, multi-document QA, summarization, few-shot learning, synthetic tasks, and code completion. We also provide per-dataset results at 25% budget in Appendix Tables 10 and 11.
>
> The analysis shows that CentroidKV performs strongly on tasks requiring semantic aggregation and reasoning, where preserving high-level contextual information is critical. In contrast, performance gaps arise in fine-grained retrieval and index-sensitive tasks, where aggressive token merging can obscure exact token matches and positional cues. Importantly, these trends are observed without any task-specific tuning, indicating that the strengths and limitations reflect inherent properties of the method. We have revised the discussion to explicitly highlight these task-dependent trade-offs and moderated our claims accordingly.
>
> ---
> **3. Tuning setup**
>
> We have added **Appendix B** to document the hyperparameters and implementation details for all methods. Baselines are evaluated using the recommended configurations from their original papers, without additional retuning.
>
> CentroidKV uses a single fixed configuration across all experiments (Section 6.1), with no task-specific calibration or dataset-dependent tuning. All results are obtained under the same setup. This ensures a fair and consistent comparison across methods and datasets.
>
> ---
> **4. Ablations for key hyperparameters**
>
> We have expanded the ablation study in **Section 6.4** to provide a more comprehensive analysis of key design parameters:
>
> - Chunk size (**Table 8**): Controls the locality of clustering (i.e., the effective clustering interval). Results show that chunking significantly reduces overhead while preserving accuracy, with $c=256$ providing the best trade-off.
>
> - Compression ratio (**Table 9**): Governs the number of retained centroids. We find that overly aggressive compression (e.g., $r_{\text{init}} = 1.0$) leads to substantial degradation, while a moderate schedule ($r_{\text{init}} = 0.80, \delta_r = 0.20$) achieves the best balance. Notably, the effective number of centroids is controlled by the compression ratio and is therefore captured through this ablation.
>
> ---
> **5. Position with respect to prior clustering-based methods**
>
> Prior clustering-based approaches (e.g., ClusterKV, ClusterAttn, IceCache) typically use clustering to guide token selection or sparse attention patterns, whereas CentroidKV directly replaces raw KV states with cluster centroids as a compressed representation. In Appendix E, we include the comparison with the representative method ClusterKV and clarify this distinction in the revised manuscript.
>
> ---
> **6. Tighten the theory-practice link**
>
> We agree that the connection between theory and implementation should be clearer. The alternating-partition optimality result primarily provides conceptual insight into the clustering behavior, rather than directly affecting implementation. To improve clarity and focus, we have moved this result to the Appendix A.

---

### Review · Reviewer_Ferb · 2026-04-03

**Summary Of Contributions:**

The paper proposes CentroidKV, a method to reduce KV-cache size in long-context LLM inference by clustering similar tokens instead of discarding them. It divides the sequence into chunks, groups similar tokens using a lightweight matching algorithm, and merges them into centroids. This reduces memory and speeds up decoding.

**Audience:**

Yes

**Audience Explanation:**

KV-cache compression is an important problem for long-context LLMs. The method is simple, efficient, and easy to apply, so it is likely useful for both researchers and practitioners.

**Broader Impact Concerns:**

There are no broader impact concerns.

**Claims And Evidence:**

Yes

**Claims Explanation:**

The experiments support the main claims, but the theory and failure cases are not fully explained.

**Requested Changes:**

- Explain failure cases where performance drops a lot (e.g., Table 2 at 25% budget where MK-NIAH-3 becomes 0.00).
- Add more analysis of important parameters like compression ratio and clustering interval i.e., currently only chunk size is studied in Table 8.
- Report latency results on more models (currently only shown for Llama in Table 6).
- Reduce strong claims like “without compromising performance” in Section 1, since some tasks clearly degrade under high compression.

---

> ### Author Response · Authors · 2026-05-07
> **Response to Reviewer Ferb**
>
> We sincerely appreciate Reviewer Ferb for the valuable comments. The concerns are addressed as follows.
>
> ---
> **1. Failure cases analysis (e.g., MK-NIAH-3 = 0.00 at 25% budget in Table 2)**
>
> We thank the reviewer for raising this point. In the revised manuscript (**Section 6.2, RULER paragraph**), we provide a more detailed analysis of tasks where CentroidKV degrades significantly. Our analysis shows that performance is closely tied to the semantic structure of tokens under cosine similarity, which is the basis of our clustering. In tasks such as MK-NIAH-1, where keys and values are semantically meaningful (e.g., natural language words or numbers), the relevant tokens remain well-separated in representation space and are thus preserved during clustering.
>
> In contrast, **MK-NIAH-3** involves randomly generated UUID-like tokens for both keys and values. These tokens lack semantic structure and exhibit weak distinguishability under cosine similarity. As a result, clustering may incorrectly merge critical tokens with surrounding context, leading to severe degradation (e.g., near-zero accuracy at aggressive compression). We have explicitly added this discussion to clarify that CentroidKV is less effective when token representations do not exhibit meaningful similarity structure.
>
> ---
> **2. Additional parameter analysis**
>
> We have expanded the ablation study in **Section 6.4** to provide a more comprehensive analysis of key design parameters:
>
> - Chunk size (**Table 8**): We evaluate different chunk sizes across context lengths from 16K to 128K. The results show that chunking effectively amortizes clustering overhead while maintaining stable accuracy. A moderate chunk size ($c=256$) provides the best trade-off between efficiency and performance.
>
> - Compression ratio (**Table 9**): We further analyze the impact of the initial compression ratio $r_{\text{init}}$ and decay rate $\delta_r$. The results indicate that a moderate schedule  ($r_{\text{init}} = 0.80, \delta_r = 0.20$) achieves a favorable balance, whereas overly aggressive compression (e.g., $r_{\text{init}} = 1.0$) leads to noticeable degradation.
>
> ---
> **3. Latency results on more models**
>
> We have added latency results on Mistral-7B-Instruct-v0.2 in the revised paper. **Table 7** presents full TTFT, TPOT, speedup, and KV cache memory comparisons across 32K–128K context lengths. The results show trends consistent with those observed on Llama: CentroidKV achieves up to $1.92\times$ decoding speedup at long context lengths, while introducing only modest TTFT overhead. This demonstrates that the efficiency benefits generalize across different model architectures.
>
> ---
> **4. Reduce strong claims**
>
> We agree that the original phrasing was overly strong and have revised the statement in Section 1 to adopt a more balanced tone:
> > "The core innovation of CentroidKV is the Chunked Soft Matching algorithm, which enables efficient KV cache clustering with a favorable accuracy–efficiency trade-off."

---

### Review · Reviewer_dvan · 2026-04-25

**Summary Of Contributions:**

CentroidKV proposes an online KV cache clustering framework for long-context LLM inference. The work is motivated by two key empirical observations.
1. Key states exhibit high, localized cosine similarity along the sequence dimension. Tokens with high similarity tend to cluster within localized regions rather than being distributed globally across the sequence. This is visualized across multiple layers and attention heads using Llama-2-7B-32K on the WikiText-2 dataset.
2. As token distance increases, the cosine similarity of key states decreases monotonically and follows a convex trend. A relationship measured over a local window of 256 tokens and averaged across attention heads.

Building on these observations, the core technical contribution is Chunked Soft Matching (CSM), a novel adaptation of Bipartite Soft Matching (BSM) from Vision Transformers to the KV cache compression problem. The key ideas are
1. Dividing key states into local chunks based on the empirical observation with cosine similarity is localized along the sequence dimension
2. Partitioning each chunk into two alternating sets to ensure similar tokens fall across sets rather than within
3. Identifying clusters via similarity-based matching and merging KV pairs into centroids using degree-weighted averaging.

The authors provide a theoretical proof of optimality for their alternating partition strategy, and significant empirical results in KV cache memory reduction, decoding speedup, and end-to-end latency reduction.

**Additional Comments:**

1. The paper is clearly written and the figure illustrating the CSM pipeline (Figure 1) is genuinely helpful. The connection to BSM from ViTs is well-motivated, appropriately credited, the challenges are identified and addressed.
2. The limitation discussion in Section 8 is honest about the static compression ratio and CPU offloading gap, which is appreciated. Framing these as future work directions is appropriate.
3. It would strengthen the paper considerably to include a qualitative analysis — for example, visualizing which tokens get merged in practice and whether the clusters are semantically coherent. This would validate the core premise more intuitively and would be compelling to a broad audience.
4. The 75% memory reduction claim requires careful interpretation. This refers to KV cache size reduction, not total GPU memory. Table 6 shows memory going from 31.83 GB to 18.36 GB at 64K context — approximately a 42% total memory reduction, which is still impressive but materially different from "75%." The distinction between KV cache memory and total GPU memory is underemphasized throughout and could mislead readers.
5. Figure 6's label says "Chelsea" instead of "CentroidKV." This appears to be a labeling error that should be corrected.

**Audience:**

Yes

**Audience Explanation:**

TL;DR: Definitely yes

The KV cache compression literature is active and growing rapidly. Researchers working on efficient LLM inference, memory-constrained deployment, and attention approximation would find several aspects genuinely interesting:

1. The theoretical result is the most academically novel contribution. Proving that the alternating partition strategy is optimal for a class of convex, monotonically decreasing score functions is a clean and portable result that could be cited by future work on token merging problems beyond KV caching.
2. The chunking as complexity reduction insight — replacing O(n²d) full pairwise similarity with O(ncd) chunk-local computation — is practically motivated and theoretically characterized. Table 1's complexity comparison is clear and useful.
3. The empirical observation about cosine similarity locality (Observation 1 and 2) is well-presented and adds the convexity characterization which is novel and useful for algorithm design.

The paper could be made more compelling to researchers who prioritize deployment realism (omission of inference frameworks like vLLM, TensorRT-LLM) or those working at model scales above 10B parameters.

**Broader Impact Concerns:**

Accuracy degradation risks in deployment

The paper focuses on average benchmark performance, but in safety-critical applications, the tail behavior under aggressive compression (25% budget) matters considerably. The worst-case degradation on tasks like MK-NIAH-3 (from 100 to near-0 for some baselines, and moderate drops for CentroidKV) warrants a discussion of when this method should not be used.

**Claims And Evidence:**

Yes

**Claims Explanation:**

TL;DR: Yes with significant caveats

Strengths:
The two motivating empirical observations (localized cosine similarity along the sequence dimension, and its monotonically decreasing convex trend with token distance) are well-supported by visualizations, and the theoretical optimality theorem is a genuine contribution. The formal grounding of the alternating partition strategy adds credibility that is often absent in systems papers of this type. The efficiency claims in Table 6 are directly measured, and the 3.19× TPOT speedup at 64K context is plausible given the quadratic nature of KV cache access.

Weaknesses in evidence:
1. Baseline comparison is limited. The paper compares against StreamingLLM, SnapKV, and PyramidKV, but omits ClusterKV, which is the most directly comparable prior work since it also uses clustering. The authors mention ClusterKV in related work and acknowledge its high computational overhead, but excluding it from experiments means the reader cannot assess the accuracy-efficiency tradeoff between the two clustering approaches. This is a significant omission.
2. Single model scale. All efficiency experiments are on Llama-3.1-8B-Instruct only. Accuracy experiments include Mistral-7B, but no model above ~8B parameters is tested. Given that the paper targets "long-context LLM inference" broadly, the absence of results on 70B-class models or MoE architectures limits generalizability claims.
3. Accuracy results are inconsistent across models. On Mistral-7B at the 25% budget (Table 5/Figure 5b), the paper quietly acknowledges that PyramidKV and SnapKV outperform CentroidKV on a non-trivial subset of tasks (TREC, TriviaQA, PassageRetrieval). The authors attribute this to "a small subset of tasks" but do not provide a rigorous statistical analysis, no confidence intervals or significance tests are reported anywhere. This is a meaningful gap given the relatively small score differences in several cells.
4. Speedup measurement methodology is narrow. All latency experiments use HuggingFace Transformers, a framework not optimized for production inference. Results on vLLM, TensorRT-LLM, or SGLang — the dominant deployment frameworks — are absent. It is unclear whether the TPOT gains survive in systems with PagedAttention or other memory management schemes.

**Requested Changes:**

1. Include ClusterKV as a baseline. This is the most directly comparable method. If its computational overhead makes it impractical, demonstrate this explicitly with runtime comparisons rather than excluding it from accuracy tables. Without this comparison, the central claim of being superior to clustering-based alternatives is unsupported.
2. Add statistical significance testing. Given the tight margins in many table cells (often 1–3 points difference), report standard deviations across runs or bootstrap confidence intervals, particularly for RULER and LongBench aggregates.
3. Evaluate on at least one larger model (≥30B parameters). The clustering overhead characteristics, the localized similarity structure, and the accuracy-compression tradeoff may differ substantially at larger scale. This is critical for the generalizability of the approach.
4. Evaluate under a production inference framework. Provide at least one experiment using vLLM or a comparable system to demonstrate that speedups persist under realistic serving conditions with memory management overhead.
5. The hyperparameter sensitivity analysis is incomplete. Table 8 evaluates chunk size but the paper does not provide sensitivity analysis over the cache ratio R, compression ratio r, or the number of attention sinks n₁ / recent tokens n₂. These are all user-specified and could meaningfully affect the accuracy-efficiency frontier.

---

> ### Author Response · Authors · 2026-05-07
> **Response to Reviewer dvan**
>
> We sincerely appreciate Reviewer dvan for the valuable comments. The concerns are addressed as follows.
>
> ---
> **1. Include ClusterKV as a baseline**
>
> We have added comparisons with ClusterKV in the revised manuscript (**Appendix E**). We would like to clarify that ClusterKV and CentroidKV differ fundamentally in design goals: ClusterKV uses clustering as a cache recall mechanism while retaining the full KV cache, whereas CentroidKV directly replaces KV states with centroid representations, enabling explicit KV cache compression and memory reduction. To make this distinction explicit, we evaluate ClusterKV using its official implementation (with custom CUDA kernels) and compare both methods against their respective full-KV baselines on Llama-3.1-8B-Instruct.
>
> As shown in **Tables 13 and 14**, ClusterKV does not reduce KV memory footprint nor improve decoding speed. In contrast, CentroidKV achieves substantial KV memory reduction and consistent decoding acceleration under the same budget. These results empirically distinguish recall-based clustering from compression-based clustering.
>
> ---
> **2. Statistical significance analysis**
>
> We have added a comprehensive statistical evaluation in **Appendix F (Tables 15–18)** using bootstrap resampling. We report mean ± standard deviation and 95\% confidence intervals for all methods across RULER and LongBench under all compression budgets and both model settings.
>
> Key findings are: on RULER, CentroidKV achieves non-overlapping confidence intervals with all baselines, indicating statistically significant improvements in aggregate performance. On LongBench (Llama-3.1-8B), CentroidKV is statistically comparable to SnapKV and PyramidKV. On LongBench (Mistral-7B), we observe statistically significant underperformance on a subset of retrieval-intensive tasks, which we explicitly acknowledge as a limitation (Section 6.2).
>
> ---
> **3. Evaluate on larger models**
>
> We agree that evaluating larger-scale models is important for generalization. Due to computational constraints, we were unable to complete experiments on 70B-class models within the revision period. However, we note that the computational complexity of Chunked Soft Matching is independent of model width, and therefore the clustering overhead does not scale with parameter count. We have explicitly identified large-scale evaluation as important future work and will include results on larger models in an extended version.
>
> ---
> **4. Evaluation under a production inference framework**
>
> We have incorporated vLLM-based evaluation results in **Appendix D (Table 12)**, integrating CentroidKV into the PagedAttention-based serving pipeline. In this setup, CentroidKV operates as a post-prefill KV compression module, where KV blocks are clustered and merged before being returned to the vLLM memory manager.
>
> Results on Llama-3.1-8B-Instruct show that CentroidKV achieves up to $1.63\times$ TPOT speedup at 128K context and $2.34\times$ –$4.0\times$ throughput improvement, depending on context length. These results demonstrate that CentroidKV’s efficiency gains persist under realistic serving conditions with memory paging and dynamic KV management.
>
> ---
> **5. Hyperparameter sensitivity analysis**
>
> We have expanded the ablation study in **Section 6.4** to improve coverage of key hyperparameters.
>
> - Chunk size (**Table 8**): We evaluate different chunk sizes across context lengths from 16K to 128K. Chunking effectively amortizes clustering overhead while maintaining stable accuracy. A moderate chunk size ($c=256$) provides the best trade-off.
>
> - Compression ratio (**Table 9**): We analyze the impact of the initial compression ratio $r_{\text{init}}$ and decay rate $\delta_r$. The results indicate that a moderate schedule  ($r_{\text{init}} = 0.80, \delta_r = 0.20$) achieves a favorable balance, whereas overly aggressive compression (e.g., $r_{\text{init}} = 1.0$) leads to noticeable degradation.
>
> In addition, we clarify that:
> - The cache ratio $R$ (75\%, 50\%, 25\%) defines the KV budget used across all experiments.
> - The number of attention sinks $n_1$ and recent tokens $n_2$ follow the recommended configuration of DuoAttention, as detailed in Appendix B.
>
> ---
> **6. Clarification of memory reduction claims**
>
> We thank the reviewer for pointing out the need for precision. We have revised the manuscript to clearly distinguish between KV cache memory reduction (Tables 6 and 7) and total GPU memory reduction.

---

### Public Comment · ~Carlos_Alberto_Matias_de_Abreu_Junior1 · 2026-06-26
**CentroidKV needs a direct analysis showing that key-based centroid merging preserves attention behavior, since downstream benchmark scores alone do not measure the internal error caused by compressing KV states.**

The revised paper provides a stronger empirical case for CentroidKV, especially after the added latency comparisons, statistical analysis, ClusterKV discussion, and clearer treatment of retrieval-heavy failure cases. However, I still think one important aspect remains under-analyzed: the paper validates the method mostly through downstream benchmark scores, but does not directly quantify the internal approximation error introduced by replacing multiple KV states with a single centroid.

The central mechanism of CentroidKV is to form clusters based on similarity in the key space and then merge both keys and values into degree-weighted centroids. This is intuitive, but it assumes that high similarity among keys is sufficient to preserve the contribution of the corresponding values for future queries. In transformer attention, keys determine retrieval, while values carry the information that is propagated. Therefore, two tokens may be close in key space while still encoding different value content. Averaging such values can create a representation that is not faithful to any original token, especially in retrieval-sensitive, index-sensitive, or rare-token scenarios.

The authors already acknowledge degradation on UUID-like and fine-grained retrieval tasks, but the current explanation remains mostly task-level. I would encourage a more mechanistic analysis that directly measures how much the compressed cache deviates from the full KV cache. For example, the paper could report: (i) the attention-output error between full attention and compressed attention across layers and heads; (ii) the relationship between key similarity and value dispersion within clusters; (iii) whether high-degree centroids accumulate larger approximation errors across decoding steps; and (iv) whether failures are concentrated in particular layers, heads, or token types.

Such an analysis would strengthen the paper substantially because it would connect the observed benchmark behavior to the actual approximation induced by CentroidKV. It would also help practitioners understand when the method is safe to use and when the cache budget should be relaxed. Without this diagnostic evidence, the method is empirically promising, but the validity of the key-based merging criterion remains only indirectly supported by downstream accuracy.

---

### Decision · Action_Editor_5Wre · 2026-05-25

**Recommendation:** Accept as is

**Audience:**

Yes

**Audience Explanation:**

KV cache compression is a widely studied and practically important problem for long-context LLM deployment. The chunked clustering approach, its theoretical grounding, and the concrete memory–latency trade-off data are directly relevant to both inference systems researchers and practitioners.

**Claims And Evidence:**

Yes

**Claims Explanation:**

All three reviewers converged on "Yes" after revision. The authors substantiated their core accuracy–efficiency trade-off claims by adding end-to-end latency comparisons against SnapKV/PyramidKV under a shared KVPress setup, vLLM-based serving evaluation, bootstrap confidence intervals, ClusterKV baselines, and expanded hyperparameter ablations. The remaining gaps — absence of evaluation beyond 8B-scale models and acknowledged degradation on retrieval-intensive tasks under aggressive compression — are explicitly scoped as limitations rather than hidden, and do not undermine the validity of the claims as stated in the revised manuscript.